

# Adaptation of the meteorological model Meso-NH to laboratory experiments: implementations and validation

Jeanne Colin[1], Christine Lac[1], Valéry Masson[1], and Alexandre Paci[1]

[1]CNRM, UMR 3589 METEO-FRANCE & CNRS. 42, av. G. Coriolis, 31057 Toulouse, France

*Correspondence to:* Jeanne Colin
(jeanne.colin@meteo.fr)

**Abstract.** A meteorological numerical model can be a powerful tool to complement laboratory experiments applied to atmospheric sciences, but it needs to be adapted in order to represent flows generated in laboratory. This paper presents such an adaptation of the atmospheric non-hydrostatic model Meso-NH. The usually neglected viscous diffusive fluxes have been added to the model's equations, along with the coding of an explicit bottom no-slip boundary condition was implemented.

These implementations are validated against exact solutions of ideal flows. Meso-NH is then used in a configuration matching a laboratory experiment performed in the CNRM large stratified water flume meant to reproduce a neutral atmospheric boundary layer. The model is run for the first time in an explicit mode (Direct Numerical Simulation – DNS) at a very high resolution (1 mm) over a large grid. The comparison with the experimental data shows that the boundary layer height and vertical profiles of mean velocity are well captured by the model. This result further validates the implementations carried out in Meso-NH

which can now be used in a DNS mode to simulate channel flows. The joint use of Meso-NH and laboratory experiments, along with the possibility to run DNS with Meso-NH, could lead to new findings or improvements in the field of atmospheric sciences.

## 1   Introduction

Classical methods used in atmospheric sciences are the analysis of observations, numerical simulations and studies based on

theoretical considerations. Laboratory experiments – in hydraulic tanks and flumes, or wind tunnels – constitute another useful means of investigation. Each method has its pros and cons, and it is often the collaboration between several approaches that leads to significant findings. Like observations, laboratory experiments provide data of the processes under scope. But contrary to Nature, the laboratory offers controlled and repeatable conditions. It also allows to isolate the various processes involved in the natural phenomenon, and/or to simplify the problem in an idealised setting.

Laboratory experiments rely on the concepts of dimensional analysis and similitude. According to the Buckingham's Π theorem (Buckingham, 1914), any complete physical relationship can be represented as one subsisting between a set of independant dimensionless quantities. Provided that these quantities remain unchanged, an atmospheric flow can be transposed to an analog in the laboratory, at a smaller scale and possibly in a different fluid. In the past, particularly from the 50s to the late 70s, laboratory experiments allowed to gain insight on atmospheric processes as various as the general atmospheric circulation



(e.g. Charba, 1974; Simpson and Britter, 1979), internal gravity waves and turbulence in stably stratified fluids (for an early review on these matters, see Thorpe (1973)), or convective boundary layers (Willis and Deardorff, 1974). In the 80s and 90s, laboratory experiments were somehow eclipsed by numerical simulations. Some features, however, remained out of reach for numerical models. Thus, there has been a regain of interest in using laboratory experiments in atmospheric sciences since the

early 2000s. Research fields using laboratory experiments include the study of internal gravity waves (e.g. Eiff and Bonneton, 2000; Knigge et al., 2010; Lacaze et al., 2013; Teixeira et al. , 2017), turbulence (Praud et al., 2006), convective entrainment (Jonker and Jiménez, 2014), stable boundary layers (e.g. Mukund et al., 2014) or interactions between wind farms and boundary layers (e.g. Porté-Agel et al., 2011; Zhang et al., 2013; Hancock and Zang, 2015). With the increase of computational ressources, it has recently become possible to study the small-scale features of such flows in Direct Numerical Simulations

(DNS). And there is now a growing body of literature based on this approach. To name a few, one can cite the work of Kimura and Herring (2012) and Almalkie and Kops (2012) on the dynamics and spectra of kinetic energy in stably stratified turbulence, or Fritts et al. (2013) who studied fine structures in gravity waves while Mellado (2012), Garcia and Mellado (2014) or Mellado et al. (2016) focused on convective boundary layer features such as the near surface properties or the entrainement zone.

In most of the aforementioned studies, the framework is either based on the analysis of experimental flows or the use of DNS. But research protocols mixing the two approaches – numerical and experimental – are also considered. For example, Jonker and Jiménez (2014) tried to reproduce the water tank experiments of Deardorff et al. (1980) which provided commonly accepted values of the buoyancy entrainment ratio. They found significantly different values of this parameter and recommend to further explore the matter with DNS (Direct Numerical Simulations). Porté-Agel et al. (2011) proposed a parametrization

of the wind turbines induced forces on the boundary layer, with the use of Large Eddy Simulations (LES) compared to wind-tunnel measurements. A broader use of numerical simulations related to laboratory experiments could be envisionned. Indeed, a numerical model can be used to complete the experimental data in various ways, provided that it is able to reproduce the flow generated in the laboratory. For example, a set of experiments can be extented with further sensitivity tests. The numerical model can also give access to a larger spatiotemporal coverage of the variables of interest, compute other variables or diagnos-

tics which can not be measured in the laboratory.

In the present study, the atmospheric research model Meso-NH (Lafore et al., 1998) is adapted to simulate experimental flows obtained in laboratory with this intent to use it as a complement to experimental data produced in laboratory. A CFD (Computational Fluid Dynamics) code such as the ones used in the aforementioned studies could very well be used for this purpose. But while CFD codes are perfectly adapted to represent experimental flows, they tend to have a rather limited spectrum of

use. Atmospheric research models like Meso-NH on the other hand are meant to simulate any atmopheric flow, ranging from the synoptic scale to the large eddy turbulent scale. The implementations presented in this study allow to run DNS (Direct Numerical Simulation) with Meso-NH. So the benefit is twofold : Meso-NH can be used as a complementary tool to laboratory experiments and its scope is extended. The possibility to run DNS with a meteorological model can be seen as an advance in itself. One of its foreseen applications is that it could offer a new framework to test parametrizations of fine-scale processes

such as the ones occurring in stable boundary layers. For example, comparing the results given by a turbulence scheme in LES



mode with those of a DNS resolving the turbulence could give new insights on the parametrization. Van Heerwaarden et al. (2017) developed a CFD code working both DNS and LES for boundary layer flows. The adaptations of Meso-NH proposed in the present study offer the same framework to the Meso-NH users community, with a more extensive range of configurations thanks to the pre-existing versatily of the model.

This was initially motivated by the existence of a geophysical fluid dynamics laboratory in our institute, which host in particular the CNRM large stratified water flume. This installation was part of the HYDRALAB European network of unique hydraulic infrastructures from 2006 to 2014. A first adaptation of Meso-NH to laboratory experiments was carried out by Gheusi et al. (2000). We started again with a more recent version of the model and performed other implementations.

Meso-NH and its adaptations are described in section 2. The two main modifications we implemented are: 1) the addition of the

viscous terms in the equations of the model, and 2) an explicit no-slip bottom condition. The reasons for these implementations are also detailed in section 2. Two types of validations are then performed: a validation against exact solution of ideal flows (section 3), and a validation against experimental data from the CNRM large stratified water flume (section 4).

By convention, in the rest of the manuscript, the word *simulation* refers to numerical simulation, whereas *experiment* stands for laboratory experiment. Depending on the case, the results will be presented directly at the scale and dimensions of the

laboratory, or the flow will be transposed into an atmospheric flow, thanks to a dimensional analysis.

## 2   The numerical model

### 2.1   Meso-NH

Meso-NH (Lafore et al., 1998) is a non-hydrostatic mesoscale research model developed by the Laboratoire d'Aérologie (UMR5560 UPS & CNRS, France) and the CNRM (UMR3589 Meteo-France & CNRS). It simulates atmospheric phenomena

ranging from the synoptic scale to the large turbulent eddy scale, and it can be applied to real situations as well as academic cases. Meso-NH is a grid point model using the Arakawa-C grid and the Gal-Chen and Somerville (1975) vertical coordinate (Figure 1). The model's dynamics is based on the anelastic approximation (filtering of the acoustic waves). The prognostic variables are the three components of the wind, the potential temperature, the mixing ratio of hydrometeors and the turbulent kinetic energy – because of the anelastic approximation, the pressure is a diagnostic variable. The numerical

schemes are an eulerian fourth-order centred advection scheme or a Weighted Essentially Non Oscillatory (WENO) scheme of fifth-order (Shu and Osher, 1988) for the momentum components associated respectively with a leapfrog or a third order Runge-Kutta time marching (Lunet et al., 2017) and the Piecewise Parabolic Method (PPM) (Collela and Woodward, 1984) advection scheme for scalar variables like temperature, associated with a forward-in-time marching. Three types of lateral boundary conditions are available: rigid wall, cyclic and open boundaries. Meso-NH can be coupled to the SURFEX surface

model (Masson et al., 2013) and it includes a large set of parametrizations. The model is freely available and can be downloaded at http://mesonh.aero.obs-mip.fr.

In the present study, we use the v5-3-0 version, and the configuration designed for academic cases with dry air. We work under the Boussinesq's hypothesis in order to take into account the incompressibility of water. This is necessary since our main





objective is to reproduce laboratory experiments using water as a fluid. Thus, we have a single phase incompressible fluid, as it is the case in a hydraulic flume. The Earth's sphericity is neglected and the horizontal coordinate system is cartesian. We also neglect the Coriolis force which does not affect the flow at the scale of a laboratory. The coupling with the surface model is disabled since we simulate laboratory experiments. No paramatrizations are activated and the implicit diffusion scheme is also

turned off. In other words, Meso-NH is run explicitly.

The anelastic approximation relies on the hypothesis that the thermodynamics variables will not depart very far from a "reference state" defined as an atmosphere at rest in hydrostatic equilibrium. A variable $V$ is therefore decomposed as $V(x,y,z,t) = V_{ref}(z) + V'(x,y,z,t)$, where $V_{ref}$ is the reference state's value of the variable and $V'$, the perturbation from this state. Meso-NH uses the Exner function $\Pi$, defined as $\Pi = (P/P_{ref})^{(R/C_p)}$ where $P$ is the pressure, $P_{ref} = 10^5$ Pa is the reference state's

pressure, $R$ the gaz constant for dry air and $C_p$ the specific heat content pressure for dry air. $\theta$, the potential temperature, is defined as $\theta = T/\Pi$ where $T$ is the air temperature.

Under the hypothesis made in our study, the model's equations are written as follows:

$$\rho' = -\rho_{ref}\frac{\theta'}{\theta_{ref}} \tag{1}$$

$$\boldsymbol{\nabla}\boldsymbol{U} = 0 \tag{2}$$

$$\frac{\partial}{\partial t}(\rho_{ref}\boldsymbol{U}) + (\boldsymbol{\nabla}.\boldsymbol{U})\boldsymbol{U} + C_p\theta_{ref}\rho_{ref}\boldsymbol{\nabla}\Pi' + \rho_{ref}\boldsymbol{g}\frac{\theta - \theta_{ref}}{\theta_{ref}} = 0 \tag{3}$$

$$\frac{\partial}{\partial t}(\rho_{ref}\theta) + \boldsymbol{\nabla}.(\rho_{ref}\theta\boldsymbol{U}) = 0 \tag{4}$$

Where $\rho$ (kg m$^{-3}$) is the air density, $\theta$ (K) and $\Pi$ (J kg$^{-1}$ K$^{-1}$), the potential temperature and the Exner function previously defined, and $\boldsymbol{U}$ (m s$^{-1}$) the wind. (1) is the equation of state, (2) the continuity equation, (3) the momentum equation and (4) the thermodynamic equation.

The following two subsections describe the implementations performed in Meso-NH to make it a suitable tool to simulate flows generated in laboratory experiments – namely, the addition of the viscous terms in the model's dynamic, and the imple-

mentation of an explicit no-slip bottom boundary condition.

## 2.2 Addition of the viscous diffusion terms

According to the Kolmogorov theory of turbulence (see Kolmogorov, 1941a, b), the turbulence kinetic energy is transfered from the largest turbulent eddies to the smallest ones until it is dissipated by viscosity. This transfer is called the energy

cascade. Kolmogorov (1941a, b) postulated that when the flow is turbulent enough, the scale of viscous dissipation – called the Kolmogorov microscale, $l_d$ (m) – depended only on the the kinematic viscosity of the fluid $\nu$ (m$^2$ s$^{-1}$) and $\epsilon$, the spectral





rate of turbulence kinetic energy transfer, which is equal to the production rate of turbulence kinetic energy. The Kolmogorov microscale $l_d$ can be expressed as:

$$l_d \approx l_e Re^{-3/4} \tag{5}$$

where $l_e$ is the scale of the largest turbulent eddies and $Re$ is the dimensionless Reynolds number. $Re$ is defined as $Re = U\Lambda/\nu$,
where $U$ (m s$^{-1}$) is the flow's velocity, $\Lambda$ (m) is a typical length of the flow and $\nu$ (m$^2$ s$^{-1}$), the kinematic viscosity. $Re$ represents the ratio of the intertial forces to viscous forces, it is commonly used to distinguish laminar and turbulent regimes – high Reynolds numbers being associated with turbulent regimes. In typical turbulent atmospheric flows, the Reynolds number reaches values of $10^7$ and above. The associated Kolmogorov microscale $l_d$ is then equal to 1 mm or less.

In meteorological models, the viscous dissipation is not taken into account. Given their typical resolution, they can not repre-
sent this phenomenon. At best (in LES), part of the turbulence remains subgrid and has to be parametrized. The dissipation of energy at the end of the spectrum is then ensured by a numerical diffusion scheme. But in our case, Meso-NH is meant to be applied to laboratory flows whose Reynolds numbers are smaller than atmospheric ones, mostly because of the reduced dimensions of laboratory installations. Typical values of $Re$ in experimental flows range from $10^2$ to $10^4$ (e.g. Eiff and Bonneton, 2000; Knigge et al., 2010). Baines and Manins (1989) have suggested that Reynolds numbers exceeding several hundred are
necessary for the flow generated in laboratory to be similar to the atmospheric one. So even though Reynolds numbers in laboratory experiments may be lower than those observed in the atmosphere, the two flows can be in a similar regime (Townsend, 1980; Wyngaard, 2010) – this is in fact closely related to the processes one studies and the critical value may depend on other parameters (see for example Brethouwer et al., 2007). In such cases, the experimental $Re$ values, associated with the smaller dimensions of laboratory, make it conceivable to explicitly resolve the turbulence up until the Kolmogorov microscale with a
numerical model, which would also represent the viscous diffusive terms of dissipation.

To do so in Meso-NH, we added the viscous diffusion terms of momemtum $F_m$ and heat $F_t$ to the momentum equation (3) and the heat equation (4), following the Navier-Stokes equations:

$$F_m \;\; = \;\; -\nu\nabla(\rho_{ref}\nabla\boldsymbol{U}) \tag{6}$$
$$F_t \;\; = \;\; -(\nu/P_r)\nabla(\rho_{ref}\nabla\theta) \tag{7}$$

where $Pr$ is the Prandtl number, defined as the ratio of the momentum diffusivity to the thermal diffusivity.

As explained in introduction, the reproduction of atmospheric flows in laboratories rests upon the concept of dimensional analysis. Conversely, an experimental flow generated in a water flume or a wind tunnel can be transposed into an equivalent atmospheric flow, provided that the dimensionless numbers characterising the flow – such as $Re$ – remain the same in both
cases. So when simulating a laboratory experiment in Meso-NH, we want to have exactly the same Reynolds number in the numerical simulation. This point raises no difficulty when the model is run directly at the scale of the laboratory, but it does when working with the dimensions of the atmosphere where the typical Reynolds number are higher. To obtain a reduced Reynolds number, one can increase the viscosity $\nu$ in the model and keep typical atmospheric values for $U$ and $\Lambda$. In the





simulated atmospheric flow, the reduced $Re$ will translate in to a bigger Kolmogorov microscale, making it also conceivable to run a DNS at the dimensions of the atmosphere.

To give an example, let us consider an atmospheric flow characterised by a mean velocity of $U = 10$ m s$^{-1}$ and a typical length of $\Lambda = 1000$ m corresponding to the size $l_e$ of the largest turbulent eddies. The viscosity of the air mainly depends on the

temperature ; for $T = 15$ °C, $\nu = 1.5 \ 10^{-5}$ m$^2$ s$^{-1}$. The Reynolds number is approximately equal to $10^9$ for this flow, and the Kolmogorov microscale $l_d$ has an order of magnitude of 0.1 mm. In a laboratory experiment, the Reynols number obtained for this flow would be smaller, due to experimental constraints – mainly on the typical length, limited by the size of the facility. Let us say it would be equal to $Re = 10^4$ as in the water flume experiments described in Lacaze et al. (2013), with a typical length of 0.1 m and a flow velocity of 0.1 m s$^{-1}$ and a water viscosity of $10^{-6}$ m$^2$ s$^{-1}$. The Kolmogorov microscale of the

experimental flow $l_d$ would be equal to 0.1 mm. Given the dimensions of the facility, it would be possible to run Meso-NH at a resolution close to $l_d$. Running the model at the atmospheric scale, if one wants to keep a velocity equal to 10 $m.s^{-1}$ and a typical length of 1000 m, the viscosity has to be equal to 1 m$^2$ s$^{-1}$, and the Kolmogorov microscale is now equal to 1 m, making it also possible to consider a DNS.

## 2.3 Implementation of an explicit no-slip bottom condition

In laboratory experiments simulating atmospheric flows in wind tunnel or water flumes, a boundary layer always develops above the ground and it interacts with the flow, as it does in the atmosphere. For example, it can have a significant effect on the orographic waves occurring in stably stratified flows. This interaction is the subject of numerous studies (e.g. Eiff and Bonneton, 2000; Vosper, 2004). The vertical development of the boundary layer can be enhanced with the use of

elements of rugosity in the experiments and reduced with the use of a smooth surface. To isolate the effects of the boundary layer, it would be necessary to suppress it in the experiments. But even with a very smooth surface, this is not possible. It can however be done with a numerical model, and this is one of the foreseen application intended to be performed with Meso-NH. Anyhow, Meso-NH must be able to simulate the boundary layers forming in laboratory experiments.

The native bottom boundary condition in Meso-NH is a "free-slip" one: the normal component of the wind is set to zero at the

ground level while no condition is applied to the tangential component of the wind. It corresponds to the following equation:

$$\boldsymbol{U}.\boldsymbol{n} = 0 \tag{8}$$

Despite this free-slip condition, Meso-NH takes into account the ground friction through the surface wind shear $u*$, expressed as a function of the roughness length $z_0$ characterising the rugosity of the surface. This formulation requires to activate a turbulence scheme and to prescribe surface turbulent fluxes. In most – if not all – simulations run with Meso-NH, these

basic requirements have raised no question so far. But in our case, Meso-NH is intended to be used explicitly to represent as realistically as possible the boundary layer close to the bottom boundary condition. Therefore, the ground friction needs to be implemented explicitly with a no-slip bottom boundary condition, reading as follows:

$$\boldsymbol{U}(z=0) = \boldsymbol{0} \tag{9}$$





This bottom boundary condition would be trivial to implement in Meso-NH if the ground level was defined for the three components of the wind. But on the Arakawa-C grid (see figure 1) of the model only $w$, the vertical component of the wind, is defined for $z = 0$. $u$ and $v$, the horizontal components of the wind, are located at the altitudes $-\Delta z/2$ and $\Delta z/2$. Physically, the no-slip bottom condition impacts the rest of the flow through the advection of momentum and the viscous diffusion of the

wind. In the free-slip condition, $w$ is already set to zero for $z = 0$. To implement the rest of the no-slip condition, the velocity of the exterior point ($z = \Delta z/2$) is set equal and opposite of that of the interior value ($z = -\Delta z/2$):

$$u(z = \frac{\Delta z}{2}) \quad = \quad -u(z = -\frac{\Delta z}{2}) \tag{10}$$

$$v(z = \frac{\Delta z}{2}) \quad = \quad -v(z = -\frac{\Delta z}{2}) \tag{11}$$

This way, the viscous diffusion of the wind $-\nu\nabla(\rho_{ref}\nabla\boldsymbol{U})$ is equal to what it would be if we could set $u$ and $v$ equal to zero

for $z = 0$. This no-slip condition is equivalent to the one defined by Gheusi et al. (2000) who set a viscous diffusion flux at the surface.

The no-slip condition can either be applied to the whole domain of integration or only to a subdomain. This feature can be necessary when for example, the flow is created by an obstacle towed in a laboratory experiment, the bottom friction is present only above the obstacle surface. Several ways of defining the subdomain on which the no-slip condition is to be applied have

been implemented. The first one is based on a condition upon the height of the surface $z_s$: the no-slip condition is applied where $z_s > z_0$ (see figure 1 for the definition of $z_s$). One can also define a subdomain with indexes in the x and y direction (rectangular subdomain), or with any given mask of points.

## 3   Validation against exact solutions of ideal flows

### 3.1   Validation of the viscous diffusion terms

We will only validate the viscous diffusion of momentum, the viscous thermal diffusion having been implemented in the same manner.

### 3.1.1   Ideal flow and analytical solution

The ideal flow we consider here is defined by the following initial state:

$$\boldsymbol{U}(x,y,z,0) \quad = \quad U_0 sin(\frac{\Pi y}{L})\boldsymbol{x} \tag{12}$$

$$P(x,y,z,0) \quad = \quad -\rho g z \tag{13}$$

$$\theta(x,y,z,0) \quad = \quad \theta_0 \tag{14}$$

where $(x,y,z)$ refers to the cartesian coordinate system, $\theta_0$ (K) and $U_0$ m s$^{-1}$) are constants and $L$ (m) is the length of the domain in the y-axis direction (the domain is infinite along the x-axis and the height z does not have to be specified for this





flow).

Neglecting the Coriolis force, the momentum equation is simplified into:

$$\frac{\partial u}{\partial t} = -\nu \frac{\partial^2 u}{\partial y^2} \qquad (15)$$

with $u = \boldsymbol{U}.\boldsymbol{x}$

With a free-slip bottom boundary condition, this equation has the following exact solution:

$$\boldsymbol{U}(x,y,z,t) = U_0 e^{\frac{-\nu \pi^2}{L^2} t} sin(\frac{\pi y}{L}) \boldsymbol{x} \qquad (16)$$

### 3.1.2 Model set-up

Meso-NH is run with dry air, with the Boussinesq approximation and without the Coriolis force and no parametrization. The initial velocity of the wind, $U_0$, is taken equal to $10$ m s$^{-1}$ and the kinematic viscosity is set at $100$ m$^2$ s$^{-1}$. With this high
viscosity, the extinction term $e^{\frac{-\nu \pi^2}{L^2} t}$ in the expression of $u$ reaches a significant value within a fairly short period of integration. In the y-direction, the lenght $L$ of the domain is $4800$ m, with rigid wall lateral boundary conditions. In the x-direction, the lateral boundary conditions are cyclic, which is equivalent to an infinite domain of integration along x. As the dimension in this direction does not matter, it consists of only 8 grid points. Since the flow is steady along the z-axis, 8 vertical levels are defined. The upper boundary condition is modelled with a rigid roof and a free-slip condition. The resolution is equal to $50$
meters in every direction, and the time step is set to $5$ s.

### 3.1.3 Results

Simulated values of $u$ (at the first vertical level) are compared with analytical solution on figure 2. $u$ is represented as a function of $y$ for $t = 0$, and after 1, 2 and 3 hours of simulation. Results show that the model is very close to the analytical solution. For the first 2 hours of simulations, the curves superimpose so well that one can barely see the difference. After 3 hours of
simulation, the difference is more visible, but the relative error stays below $2.7\%$.

From this validation, we conclude that our implementation of the viscous terms is correct.

### 3.2 1D Validation of the explicit no-slip bottom condition: the first Stokes problem

### 3.2.1 Ideal flow and analytical solution

This flow is known as the "Stokes's first problem" (Schlichting, 1968). It occurs over a suddenly accelerated flat plate. Over
time, a boundary layer develops above the plate and it can be analytically described.

With an infinite bottom surface started impulsively from rest to a speed of $U_0$ in the x-axis direction, we obtain the simplified momentum equation:

$$\frac{\partial u}{\partial t} = -\nu \frac{\partial^2 u}{\partial z^2} \qquad (17)$$





with the following boundary conditions:

$$t \leq 0 \quad : \quad u = 0 \tag{18}$$

$$t > 0 \quad : \quad u = U_0 \text{ for } z = 0 \text{ and } u = 0 \text{ for } z = \infty \tag{19}$$

If we define its height $\delta$ as the height for which $u(\delta) = 0.01$, we obtain:

$$\delta \approx 3.64\sqrt{\nu t} \tag{20}$$

See Schlichting (1968) for a demonstration of this result.

This problem is equivalent to the one where the fluid would be set at the uniform speed of $U_0$ , with a free-slip on the bottom surface, and then suddenly submitted to the ground friction. In this case, the momentum equation is the same as (16) and the boundary conditions become:

$$t \leq 0 \quad : \quad u = U_0 \tag{21}$$

$$t > 0 \quad : \quad u = 0 \text{ for } z = 0 \text{ and } u = U_0 \text{ for } z = \infty \tag{22}$$

With a boundary layer thickness $\delta$ defined as $u(\delta) = 0.99\,U_0$, we still have $\delta \approx 3.64\sqrt{\nu t}$.

In Meso-NH, the ground can not move. But this sudden shift from a free-slip to a no-slip bottom condition is exactly what happens when the model starts from an initial state where the velocity is set at a uniform value over the whole domain of integration.

### 3.2.2 Model set-up

The initial wind velocity, $U_0$, is taken equal to $10$ m s$^{-1}$. Three different values of viscosity are considered:

- $\nu_1 = 0.5 \; m^2.s^{-1}$,

- $\nu_2 = 0.1 \; m^2.s^{-1}$,

- $\nu_3 = 0.064 \; m^2.s^{-1}$

These values of viscosities match the range of values obtained for typical Reynolds numbers in laboratory experiments ($\nu_3$ is equal to the kinematic viscosity that will be defined in section 4 for the validation against experimental data).

Here again, Meso-NH is used explicitly. It is run for $1000$ s, with the fourth-order centred scheme to transport momentum, inducing a time step of $0.5$ s. Since the flat ground is infinite, the flow is essentially one dimensional. In the model, this feature is obtained with cyclic boundary conditions in both horizontal directions. Thus, the horizontal size of the domain does not matter, and the resolution is not crucial either. 10 points are defined in each direction and the horizontal resolution is equal to $1$ m. In the ideal flow, the velocity in the free stream above the boundary layer remains equal to $U_0$. In the model however, the decrease of velocity in the boundary layer will be compensated by an increase in the upper levels for conservation purposes. To limit the impact of this effect on the comparison to the ideal flow, the domain of integration must be high enough. It is set



to 2000 m. The vertical resolution is equal to 1 m in the first hundred meters where, according to equation (19), the boundary layer is expected to develop during a 1000-second long simulation. Above, the resolution ranges from 2 m to 10 m, and it is equal to 50 m in the upper levels.

### 3.2.3 Results

5   The results are shown on figure 3, where the simulated and analytical values of $\delta$, the boundary layer height, are plotted against $t$, the time of integration. In all three simulations, the model's values of $\delta$ stay very close to the ones analytically computed. The differences are slighty higher in the third simulation, where the kinematic viscosity is the smallest ($\nu = 0.064 \text{ m}^2 \text{ s}^{-1}$), which could indicate that the resolution of 1 m is not quite high enough for this case. But even so, the solution provided by the model stays very close to the analytical one, with relative differences smaller than 5 %.

## 3.3 2D Validation of the explicit no-slip bottom condition: the Blasius boundary layer

### 3.3.1 Ideal flow and analytical solution

The Blasius theory (Blasius, 1908) refers to a boundary layer developing over a semi-infinite smooth plate in the absence of turbulence. It provides an analytical solution for the boundary layer height $\delta_0$ (defined as in section 3.2) as a function of $x$ (the 15   x-axis being the flow's direction) in permanent regime:

$$\delta_0 \approx 5\sqrt{\frac{\nu x}{u_\infty}} \tag{23}$$

### 3.3.2 Model set-up

As in the previous sections, the initial velocity of the wind, $U_0$ is set at 10 m.s$^{-1}$, the viscosity $\nu$ is taken equal to $0.064 \text{ m}^2 \text{ s}^{-1}$ and Meso-NH is run explicitly. In the flow's direction $(x)$, the domain is 20 km long with a resolution of 200 m – here again, 20   the resolution does not really matter – with open boundary condition on the outflow. In the $(y)$ direction, there is only 5 points, with the same resolution of 200 $m$ and rigid wall boundary conditions. For the same reason as in the previous section, the domain of integration must be high enough for the simulation to be comparable to the ideal flow. It is set at $10,000$ m, with a resolution ranging from 1 m in the boundary layer to 2000 m in the upper levels. The model is run for 2 hours, with a time step of 5 s.

25  ### 3.3.3 Results

Figure 4 shows the analytical boundary layer height along (X) and the one simulated by Meso-NH. Both are very close. The small differences can be explained by the fact the velocity of the free flow is not exactly equal to $U_0$ everywhere in the domain as it is in the ideal flow.





The results of this section validate our implementation of the viscous diffusion and the no-slip bottom condition in the case of an ideal laminar flow. However, it is not sufficient to conclude that this implementation allows to realistically simulate boundary layers forming above smooth surfaces in laboratory experiments. This point is addressed in the following section.

## 4    Validation against experimental data

### 4.1    The CNRM water flume

The experiment used in this section has been conducted in a large stratified water flume (see picture on figure 5) which is the main installation at the geophysical fluid dynamics laboratory of the CNRM. Built in 1982, this flume was initially designed for applied studies of atmospheric flows over complex terrain. It soon became used for research purposes, in particular for studies on internal waves and boundary layers (for recent studies see Knigge et al., 2010; Lacaze et al., 2013; Dossman et al., 2014).

It is a 22-meter long, 3-meter wide and 1-meter deep water flume that can operate with up to three layers of different densities (NaCl brines ranging from 1000 to 1200 kg m$^{-3}$ monitored by computer), and velocities (ranging from 0.03 to 0.75 m s$^{-1}$). The flume can also be operated as a towing tank filled with water for homogeneous flows or with density-stratified brines of any stable vertical profile. The towing speed ranges from 0.01 to 0.8 m s$^{-1}$. Large and heavy obstacles are easily towed and the instrumentation can be transported on the carriage.

This facility is unique in particular due to its capacity to generate high-Reynolds numbers stratified flows with low confinement effects. The range of heights of the boundary layers typically developing in this facility allow the use of reduced size models with scales ranging from $1 : 50$ to $1 : 10000$.

The laboratory also includes two smaller water tanks (4-meter and 7-meter long) as well as a 2.5 meters rotating turntable. The research activities of the laboratory ended at the end of 2014 due to severe budget cut, but part of the equipment is kept in use 20   for lectures.

### 4.2    Description of the experimental data

Figure 6 presents a sketch of the experiment used here to evaluate the modification implemented in Meso-NH. This experiment has been chosen in order to isolate the effects of the ground friction from other processes and make a proper evaluation of the numerical code. In this rather old experiment (Perrier and Butet, 1988), the CNRM large stratified water flume was used with a 25   1-meter deep single layer of water (no density stratification) circulating at a velocity of 25 cm s$^{-1}$. The relative mean velocity fluctuation was less than 1 %, thanks to the controlled stability of the pump regime $(1/1000)$ and for the water temperature (less than 0.1 °C per day).

The first five meters of the flume (on the right on figure 6) are used to laminarize the flow coming out of the pumps and pipes located under the canal. It is composed of three grids with a decreasing mesh size going from 20 mm to 0.5 mm. At the outflow 30   of this 5-meter chamber, the turbulence intensity is less than 2 % – to be compared to the 20 % measured at the entrance of the flume. The next 17 meters of the flume are composed of a simple flat and smooth floor, over which a turbulent boundary layer





develops. It typically reaches about 10 cm, so the Reynolds number is about 25000.

The horizontal velocity measurements were performed with TSI 1218-20 W hot-film probes and TSI IFA-100 constant temperature anemometer. The probes were placed on a 3D moving bench – with a precision of $0.1$ mm. Values are presented in the (0,x,y,z) cartesian coordinate system, where the x-axis is parallel to the flow and the origin point is centred on the median axis

of the canal, on the floor just downstream of the five meters laminarizing chamber. Vertical profiles of mean velocity, $\overline{u}(z)$, were performed close to the median axis (at $y = 30$ cm), with x ranging from $1.5$ to $12$ meters, with a point of measurement every $1.5$ m and a vertical resolution ranging from $0.1$ cm near the ground to $2$ cm near the surface. The flow was sampled at a frequency of $50$ Hz, each point representing between $1000$ and $3000$ measurements.

The vertical profiles of $\overline{u}$ show an increase of velocities at the vicinity of the floor, with a maximum $\overline{u}_{max}$ superior to $\overline{u}_{\infty}$, the

mean velocity above the boundary layer (see figure 7). This is due to the experimental configuration where the floor is actually a false floor. This feature of $\overline{u}(z)$ is more pronounced for the profiles close to the entrance of the flow and it almost disappears for $x \geq 9\ m$. It makes it difficult to define a boundary layer height that would be consistent along the x-axis. In the previous sections, we used a boundary layer thickness $\delta_0$ defined as:

$$\overline{u}(\delta) = 0.99\,\overline{u}_{\infty} \tag{24}$$

But in the present laboratory experiment, when the vertical profiles of $\overline{u}$ show a significant increase of velocity near the floor, this classic definition does not make sens anymore. To overcome this problem, two other criteria were defined by Perrier and Butet (1988). These are $\delta_1$ and $\delta_2$, defined as:

$$\sqrt{\overline{u'^2}}/\overline{u}(\delta_1) = 0.01 \tag{25}$$

$$\overline{u}(\delta_2) = \overline{u}_{max} \tag{26}$$

For $x \geq 7.5\ m$, we were able to define $\delta_0$ as $u(\delta_0) = 0.99\overline{u}_{\infty}$ by considering the values of $\overline{u}$ for $z > \delta_2$. These three definitions of $\delta$ are illustrated on figure 7. Wherever $\delta_0$ can not be defined because of the false floor effect, it seems reasonable to assume that if this effect was suppressed, the value of $\delta_0$ would lie somewhere between $\delta_1$ and $\delta_2$. Therefore, we will use $\delta_0 = 0.5(\delta_1 + \delta_2)$ in these cases.

### 4.3 Model set-up and numerical simulations

As explained in introduction, a flow generated in a hydraulic flume can be transposed into an analog atmospheric flow. To do so, one has to list all the independant parameters of the experimental flow. Then, dimensionless numbers characterising the problem can be defined. Their number is given by the Vaschy-Buckingham theorem which states that a problem with $p$ independant parameters and $n$ dimensions can be fully described with $p - n$ dimensionless numbers. Here, the independant parameters of the flow are: $L$, the length of the flume ; $W$, its width ; $H$, the height of the fluid's layer ; $\nu$, the fluid's kinematic

viscosity ; $U$, its velocity ; $\Lambda$, a typical length characterising the flow ; and $g$, the gravitational acceleration. There are six independant variables and two dimensions (length and time). So four dimensionless numbers are necessary to fully describe the flow. We consider:

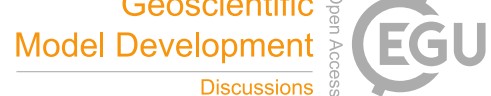

- $R_e = U.\Lambda/\nu$

- $F_r = U/\sqrt{gH}$

- $d_1 = L/H$

- $d_2 = L/W$

As long as these numbers have the same value in the atmospheric flow and the experimental one, both problems are equivalent. To scale the problem at atmospheric dimensions, we take $L_a = 1600\ L_f$, where the subscript $a$ refers to the atmospheric flow and $f$ refers to the experimental flow in the flume. From this scaling and the invariance of the dimensionless numbers, we can deduce the values of all the parameters for Meso-NH. These are listed in table 1. Given this equivalence, the model can either be run at the scale and dimensions of the flume or at the corresponding atmospheric dimensions. In the previous section,

Meso-NH was run with dimensions typical to those of the atmosphere. Here, on the contrary, the model is run at the scale of the laboratory, all results will be presented accordingly. We just provide the corresponding atmospheric dimensions because they are more meaningful to atmosphere modellers (see table 2).

Our goal is to simulate the experimental flow with a DNS (Direct Numerical Simulation) that would accurately represent the observed turbulence, without any parametrization. We performed such a simulation – that we will simply call **DNS** – with a

horizontal resolution of 1 mm and a vertical resolution ranging from 1 mm in the boundary layer to 10 mm at the top. The initial velocity $U_0$ is set at an uniform value of 0.25 m s$^{-1}$. The momentum is advected with the WENO 5th order scheme associated with a 3rd order Explicit Runge-Kutta temporal scheme. This DNS simulation is run for 82 seconds in order to reach stationarity, with a time step of 5 10$^{-3}$ s. Given the high computational cost of this simulation, the integration domain was reduced compared to the flume's true dimensions. It is 12.5 meters long with an open boundary condition in the $(x)$ direction,

0.2 meters wide with a rigid wall boundary condition in (y), and 0.5 m high and a rigid roof condition at the top. Therefore the computational domain includes 12500 x 200 x 200 points. The flow being essentially bidimensional along the $(x)$ and $(z)$ directions, it did not appear crucial to represent the full width of the flume. And since the boundary condition is open in the $(x)$ direction, it is not necessary to represent the total length of the flume either, but only to cover the portion in which the measurements were made. In the z-direction, a sponge zone is defined in the last 0.1 m at the top, where a Rayleigh damping

is applied. The no-slip bottom condition is applied over the whole domain, starting from $x = 0.1$ m to mimic the effects of the false floor and avoid boundary conditions problems at the entrance of the flow. In the laboratory experiment, the ground is not perfectly smooth and the velocity imposed at the entrance of the flow is not perfectly constant and equal to $U_0$ over time and space. These imperfections constitute two sources of turbulence. In order to represent them in the model, a white noise is applied on the three component of the initial velocity $U_0$, and the same white noise is then added at every time step on the

velocity $U$ of the inflow and the outflow ($x = 0$ m and $x = 12.5$ m).

Ideally, in order to accurately simulate the experimental flow in an explicit simulation, the resolution should be close to the Kolmogorov microscale $_d$ defined by equation 5 ($l_d \approx l_e Re^{-3/4}$, with $l_e$, the scale of the largest structures, and $Re$, the Reynolds number). Here, we have $R_e = 25000$, if we consider $l_e$ to be equal to $\delta$, the boundary layer thickness, it gives





$l_d \approx 5 \; 10^{-5}$ m at the scale of the flume ($l_d \approx 0.1$ m at the scale of the atmosphere). With our current computational ressources, it is not possible to run Meso-NH at this resolution. However according to Yakovenko et al. (2011), it is possible to run **DNS** for resolutions which are around 10 times the Kolmogorov microscale. We argue – and results will confirm – that the resolution of 1 mm, which is 20 times the Kolmogorov microscale, is sufficient.

## 4.4 Results

An overview of the simulated flow is presented on figures 8 and 9 with a vertical slice of the velocitie $U$ along the (x) direction, and several three-dimensional views of $U$. They were computed with instantaneous outputs at the end of the simulations ($t = 82$ s). These plots show the thickening of the boundary layer along $x$ and the turbulent nature of the simulated flow. They reveal small-scale 3D isotropic patterns of turbulence in the boundary layer. This indicates that our version of Meso-NH might
indeed be able to resolve the whole spectrum of turbulence for this flow. To determine if this is actually the case, we will now compare the **DNS** results to the laboratory measurements in terms of boundary layer heights, vertical profiles of velocity and turbulence intensities, and other characteristics of the turbulence.

To compute further diagnostics in **DNS**, instantaneous velocities will need to be averaged over time in stationary regime.
A statistical test is performed to determine the time $t_s$ when the flow becomes stationary. We consider $u$ the x-component of the velocity for $x = 12$ m and $y = 0.1$ m (at the outflow, in the middle of the flume), averaged over the last $0.5$ seconds at a given time step. For each time step $t$, we compute an estimate of the tendency between the inital time $t$ and the last time step of the simulation. Then a Student test is performed on the hypothesis "the tendency is null" (hence, the flow has reached a stationary regime). P-values of the test, computed for each time step, are plotted on figure 10 (low p-values correspond to a
high confidence in this hypothesis). The p-values start dropping at $t \approx 40s$, and with a confidence level of 5 %, the stationary regime begins at $t = 59s$. So when necessary, the quantities will be averaged from $t = 59$ s to the end of the simulation at $t = 82$ s.

The mean boundary layer height $\delta_0$ (see section 4.2) is computed and compared to the experimental data and the Blasius
analytical solution. The results are shown on figure 11 where $\delta_0$ is plotted against $x$. To give an idea of the uncertainties on this parameter, we also plotted the observed values of $\delta_1$ and $\delta_2$, as well as the simulated $\delta_1$ (there is no $\delta_2$ in the simulation). **DNS** stays fairly close to the experimental data. It follows almost exactly the observed values in the first 2 meters. Up until $x = 6$ m, the simulated boundary layer stays within the range of uncertainty given by $\delta_1$ and $\delta_2$, after which the model tends to underestimate the height of the boundary layer. Overall, with errors staying below 18 % on $\delta_0$, one can say that **DNS** provides
rather realistic values of the boundary layer height.

We will now examine vertical profiles of $\overline{u}$ and turbulence intensities of **DNS** for a further assessment. Figure 12 compares the simulated profiles $\overline{u}$ for the eight values of $x$ where the measurements were performed. For the first two profiles ($x = 1.5$ m and $x = 3$ m), the simulated profiles do not compare very well to the experimental data. For these values of $x$, the experimental



flow is still quite influenced by the false floor effect causing a strong increase of $\overline{u}$ near the floor in the data, which is not reproduced in the simulation. As this unwanted feature weakens along the x-axis, the simulated and observed profiles become more comparable. The simulated velocities prove to be quite accurate in the boundary layer, while $\overline{u}$ becomes too strong above the boundary layer. Arguably, this error in **DNS** could have been reduced if the damping zone (starting at $z = 0.4$ m) had been placed higher. But this would have required to define a bigger domain of integration which would have increased the computational cost. We believe that the results of **DNS** are realistic enough as they are for the intended validation purposes of our implementations in Meso-NH.

Along with mean velocity profiles, the turbulence intensity $I$ ($I = \sqrt{\overline{u'^2}}/\overline{u}$) was also measured in the laboratory experiments. In the **DNS** simulation, these quantities were directly computed with the instantaneous outputs and the values of velocity averaged over a 5-second time period. The comparison is presentend on figure 13. It shows that the simulated profiles of $I$ are quite realistic in the boundary layer. Above, the turbulence of **DNS** is virtually null. This is not surprising, as the two sources of turbulence are the white noise applied on the inflow (for $x = 0$ m) and the vertical sheer of velocity which is present only in the boundary layer.

The von Karman and Prandtl theory gives a universal law for $\overline{u}$ in the surface boundary layer in neutral conditions of stratification:

$$\overline{u}(z) = \frac{u*}{\kappa} ln(\frac{z}{z_0}) \tag{27}$$

with $\kappa$, the universal constant of Karman $\kappa \approx 0.41$ ; $u* = \sqrt{\overline{u'w'}}$, the mean turbulent flux and $z_0$, the roughness length characterising the rugosity of the ground. In situ observations in the atmosphere follow this universal profile. It is valid for $z >> z_0$ and $z << \delta_0$, that is to say not too close to the ground or the top of the boundary layer.

The values of $u*$ along $x$ were measured in the laboratory experiment and they can be computed in the model. To do so, we proceeded as for $I$, with the instantaneous and 5-second averages of the horizontal and vertical velocities. The results are given in table 3. The experimental and simulated values are similar, providing a more objective validation of the turbulence characteristics in **DNS**.

In the model where the ground is perfectly smooth, it is not possible to give a theoretical value of $z_0$. However, a logarithmic regression can be performed on the vertical profiles of velocity, from which estimates of $u*$ and $z_0$ can be deduced – they will be refered to as $\widehat{u*}$ and $\widehat{z_0}$. We did so in **DNS** and for the experimental data, for values of $z$ comprised between $\delta_0/10$ and $\delta_0/2$. The regression coefficient obtained are all above 0.97, indicating that the profiles are indeed logarithmic in this portion of the boundary layer, as one can see on figure 14. Furthermore, in most cases, the slopes of the regression are quite similar in the model and the experimental data. Table 4 gives the values of $\widehat{u*}$ and $\widehat{z_0}$ deduced from the regressions and the universal law of the von Karman and Prandtl theory. The fitted values $\widehat{u*}$ (table 4) compare rather well with the values of $u*$ computed in the **DNS** or measured in the experimental flow (table 3). It is not the case for the first two experimental profiles where the flow



is disturbed by the false floor effect but for these two profiles, the logarithmic regression does not make much sense anyway. Elsewhere, $\widehat{u*}$ tends to be slightly higher than $u*$, in both **DNS** and the data. This means that the simulated and experimental profiles of velocity in the boundary layer are steeper than what the von Karman and Prandtl theory predicts. Regarding $z_0$, the values deduced from the logarithmic regression have the same order of magnitude in the experimental data and in **DNS** (again,
the first two profiles are discarded because of the false floor effect).

To sum up, results showed that Meso-NH is able to accurately simulate the turbulent flow observed in the laboratory. With the implementations presented in this study, Meso-NH has thus proven to be successful in explicitly representing the turbulence developing in laboratory experiments. With a computational domain of half a billion grid points, this **DNS** was made possible
thanks to the adaptation of Meso-NH for massively parallel computing (Pantillon et al., 2011).

## 5 Conclusions

In this paper, we have presented an adaptation of the numerical meteorological model Meso-NH dedicated to the representation of experimental flows generated in hydraulic flumes or wind tunnels. In the same way atmospheric flows can be physically simulated in hydraulic flumes, these laboratory experiments can be conversely represented in an atmospheric model. The idea
is to use this model as a tool to extend or complete laboratory experiments in various ways such as extending a set of sensitivity experiments or giving access to a larger spatiotemporal cover of variables.

The implementations that were performed in the Meso-NH code for this purpose are: (1) the addition of the viscous diffusion terms in the equation of the momentum and in the heat equation, and (2) an explicit no-slip bottom condition. The viscous diffusion terms were added to represent explicitly the viscosity in laboratory experiments and to resolve the viscous dissipation
at the end of the turbulence energy spectrum, which is necessary to run Direct Numerical Simulations (DNS). With this first implementation, it becomes possible to explicitly represent the bottom boundary condition of no-slip – otherwise simulated through the paramatrization of the turbulence. Thanks to these two implementations, Meso-NH can be used to simulate experimental flows generated in hydraulic flumes or wind tunnels.

First, we validated our implementations in idealised configurations where the equations describing the flow have an analytical
solution. The model outputs were compared to the theory. Then, our modified version of Meso-NH was used to simulate an experimental flow generated in simple conditions in the CNRM hydraulic flume. A DNS was run at a very high resolution (1 mm) and without any parametrization. It reproduced quite well the turbulent flow obtained in laboratory. To our knowledge, this is the first time an atmospheric model is successfully run in DNS. Of course, it would be far too expensive to run Meso-NH in DNS mode at the typical dimensions of the atmosphere. This configuration of the model is meant to be a complementary tool
to laboratory experiments. Unfortunately, due to budget cuts, the CNRM hydraulic flume was closed down, thus reducing the short-term prospects of this new numerical tool. However, there are still plans to use this tool in complement to past experiments conducted in this flume such as the one descrbied in Stiperski et al. (2017). Moreover, other facilities conducting laboratory experiments are free to use our version of Meso-NH which can be downloaded on the website http://mesonh.aero.obs-mip.fr.





Nonetheless, the DNS simulation we performed in this study can open some new prospects in the development of Meso-NH for the future. For instance, equivalent DNS simulations could be used as references to assess the parametrization of the turbulence based on the prognostic equation for the turbulence kinetic energy (TKE) with a closure by mixing length (Cuxart et al., 2000) in Meso-NH. A new mixing length (Rodier et al., 2017) has been recently implemented in the model for neutral stratification. It

could be evaluated with the DNS in terms of turbulent intensity and eddies size. Hence, a DNS configuration could prove very useful to improve and validate turbulence schemes in LES, the same way LES simulations have been used for years to develop and improve convection paramatrizations in GCM (General Circulation Models) and NWP (Numerical Weather Prediction) (see for example Siebesma et al., 2003; Rio and Hourdin, 2008). In any case, we believe the possibility to run Meso-NH in a DNS mode constitutes a noteworthy advance in the model's development.

*Code availability.*    The model is freely available and can be downloaded at http://mesonh.aero.obs-mip.fr/mesonh53/Download

*Competing interests.*    The authors declare that they have no conflict of interest.

*Acknowledgements.*    We thank J. Stein, N. Merlet, F. Stoop and F. Auclair for their contributions to the laboratory experiment version of Meso-NH project which started several years ago at the CNRM Geophysical Fluid Mechanics Laboratory in Toulouse. We thank O. Thual for a helpful discussion regarding the ideal flow used in section 3.1. Experiments used in section 4 have been led by M. Perrier and A. Butet

from the CNRM Geophysical Fluid Mechanics Laboratory in Toulouse. We thank Philippe Wautelet for the 3D figures. Finally we thank J-C. Canonici who designed the original scheme from which figure 6 was designed.





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

1G



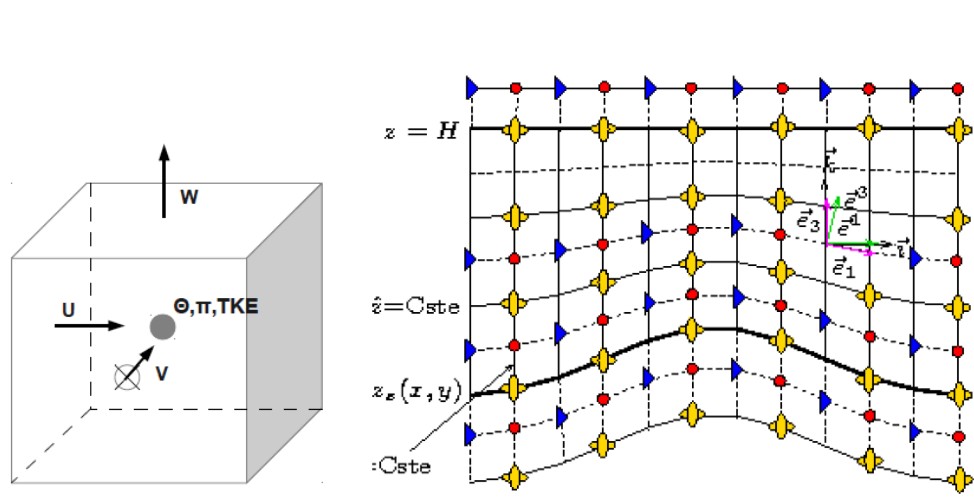

(a)                                    (b)

**Figure 1.** *Meso-NH grid. (a): Arakawa-C grid ; (b): Gal-Chen and Somerville (1975) vertical coordinate.*





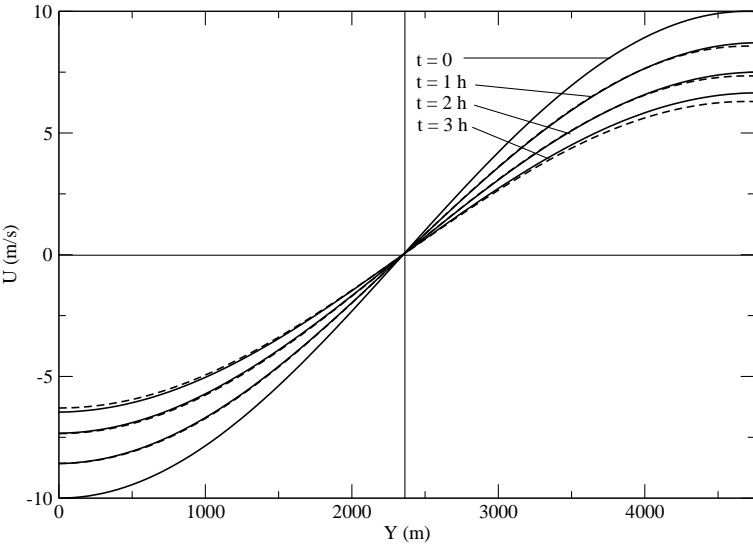

**Figure 2.** *Velocity of the flow $u(y)$ ($m.s^{-1}$) defined by equations (12) to (16) for $t = 0\ s$ ; $t = 1\ h$ ; $t = 2\ h$ and $t = 3\ h$. Solid lines: velocity simulated by Meso-NH ; dashed lines: velocity given by the analytical solution (equation (16)).*

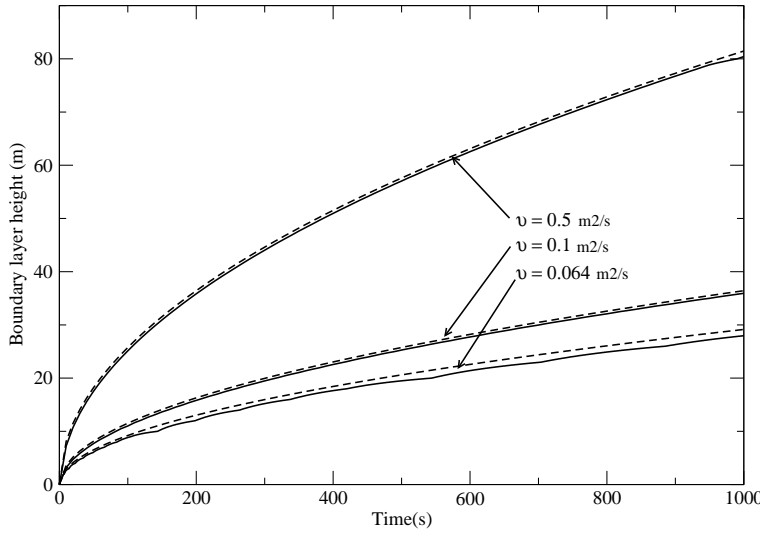

**Figure 3.** *Development of the boundary layer height in the "First Stokes's problem". Height of the boundary layer, $\delta$ (m), over time (s) for three values of kinematic viscosity: $\nu = 0.5\ m^2.s^{-1}$ ; $\nu = 0.1\ m^2\ s^{-1}$ and $\nu = 0.064\ m^2\ s^{-1}$. Solid lines: $\delta$ simulated by Meso-NH ; dashed lines: $\delta$ given by the analytical solution (equation (20)).*





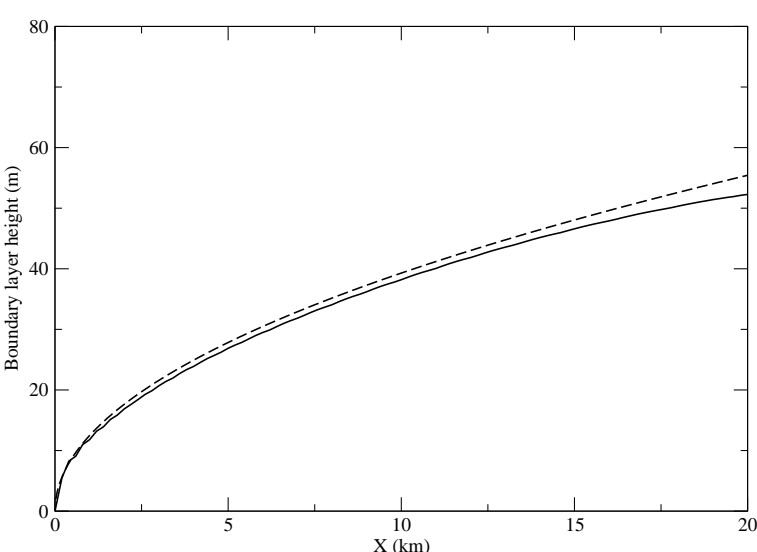

**Figure 4.** *Development of the Blasius boundary layer. Height of the boundary layer δ (m) along X (km). Solid line: δ simulated by Meso-NH ; dashed lines: δ given by the analytical solution (equation (23)).*



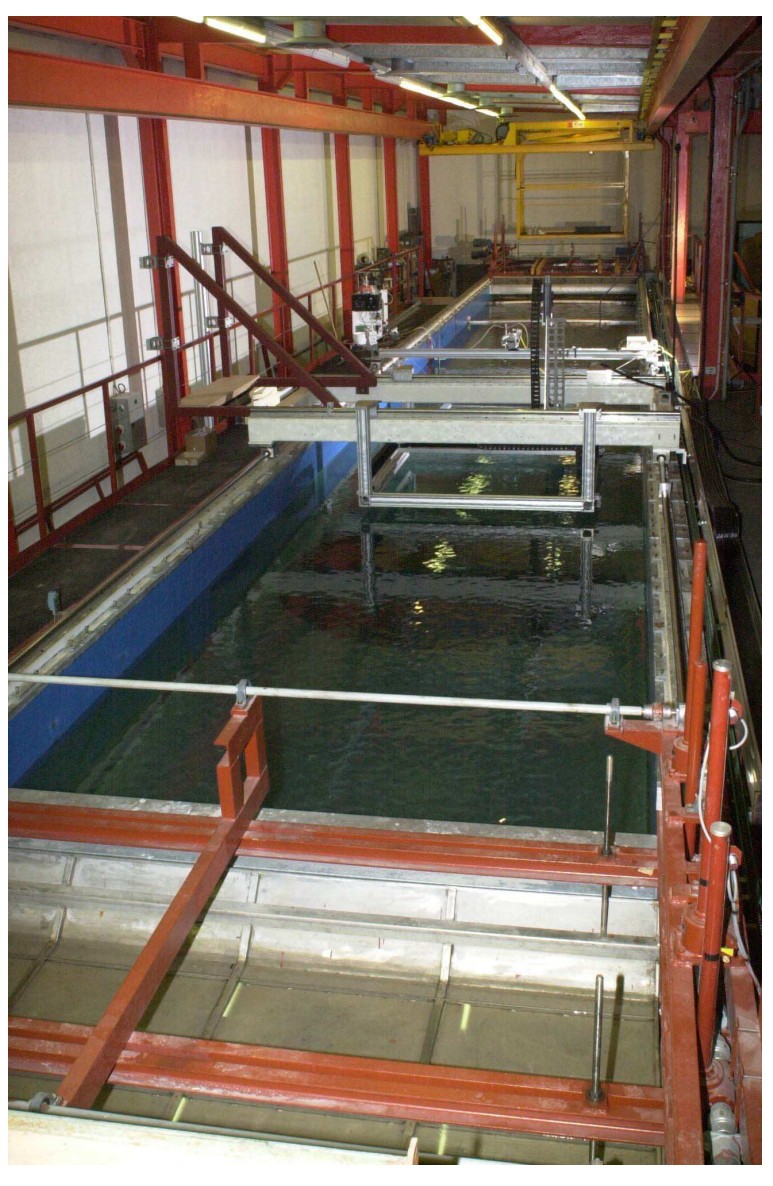

**Figure 5.** *View of the CNRM water flume.*





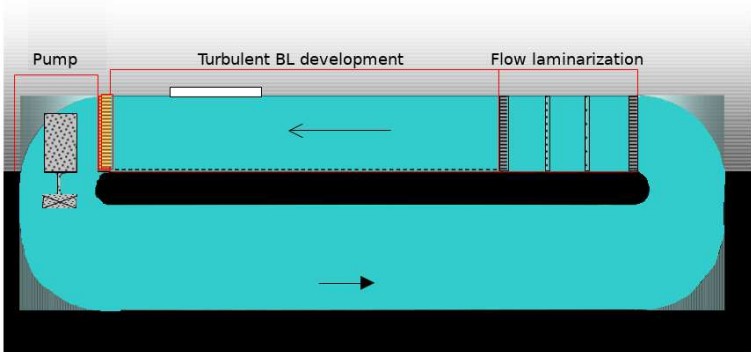

**Figure 6.** *Sketch of the experiment in the CNRM large stratified water flume used to evaluate the numerical simulation.*

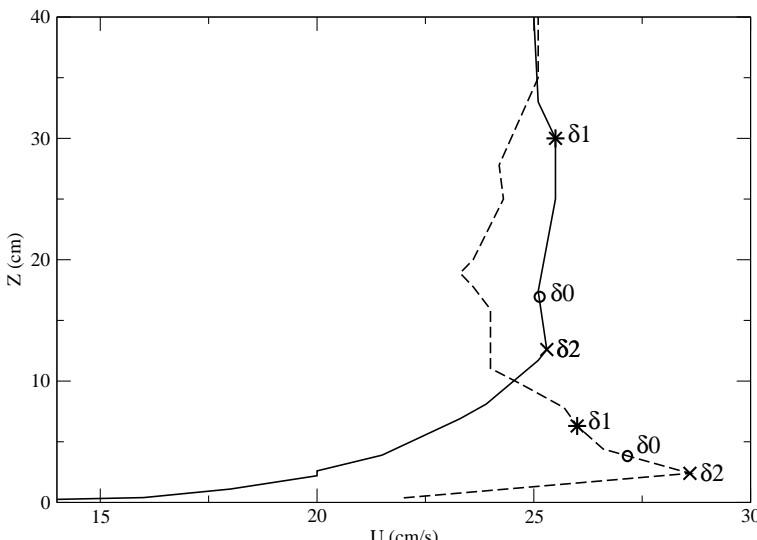

**Figure 7.** *Criteria for the boundary layer height $\delta_0$, $\delta_1$ and $\delta_2$ (cm) (equations (24), (25) and (26)) on measured vertical profiles of mean velocity $\overline{u}(z)$ (cm s$^{-1}$). Dashed line: $\overline{u}(z)$ for $x = 1.5\ m$ ; solid line: $\overline{u}(z)$ for $x = 12\ m$.*





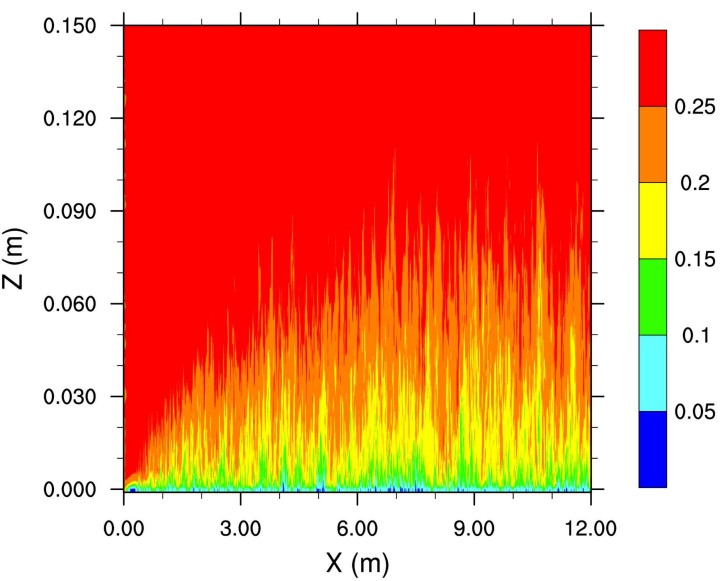

**Figure 8.** *Vertical slice of mean velocities (m.s$^{-1}$) in the (x) direction at the center of the flume in (y) and at the end of the simulation (t = 82 s).*

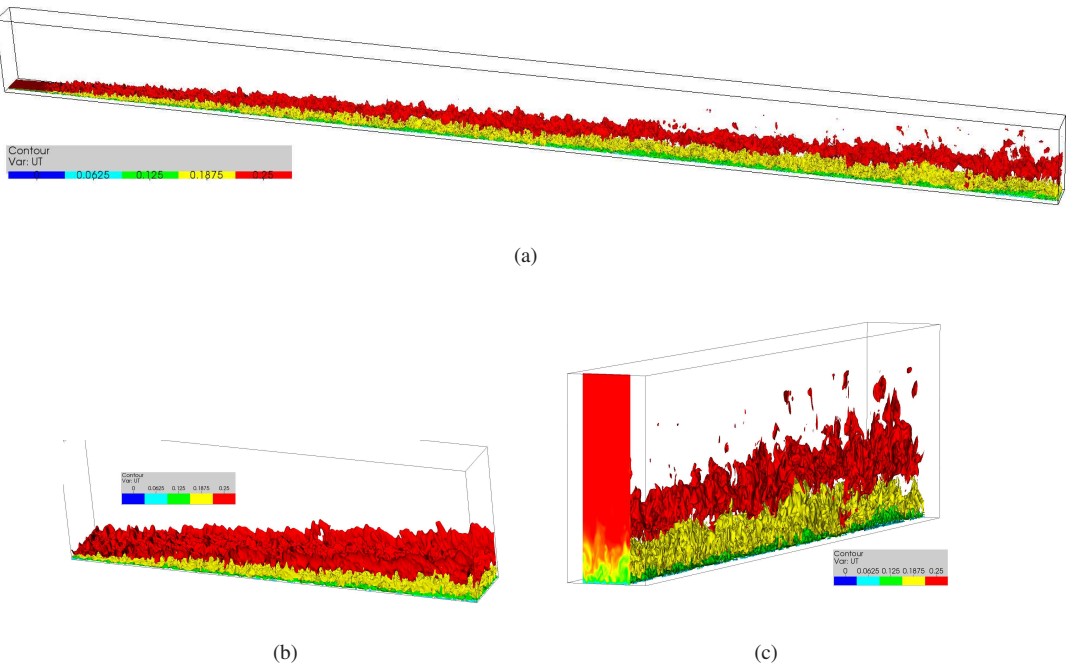

**Figure 9.** *3D views at the center of the flow (y = 4 cm to y = 16 cm). (a): overall view for x = 0.5 cm to x = 7 m ; (b):view at the entrance of the flow for x = 0.5 cm to x = 2 m ; (c): view in the middle, for x = 5 m to x = 7 m.*





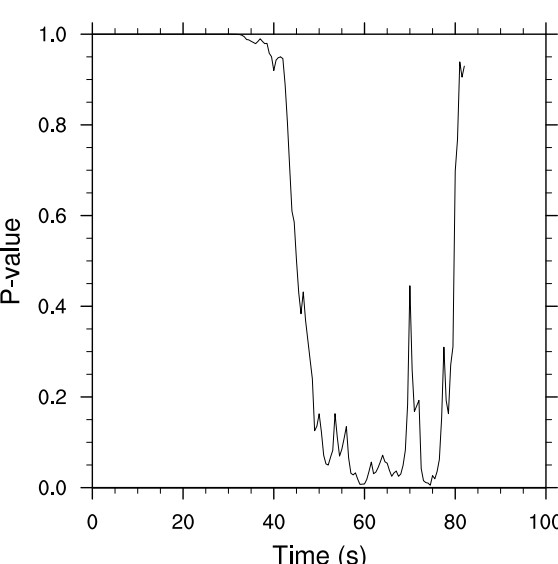

**Figure 10.** *P-values of the Student test testing the hypothesis "The tendency of the mean velocity at the outflow between $t = 0$ and $t$ is equal to zero", $t$ being the time, varying from $t = 0$ s to $t = 82$ s.*





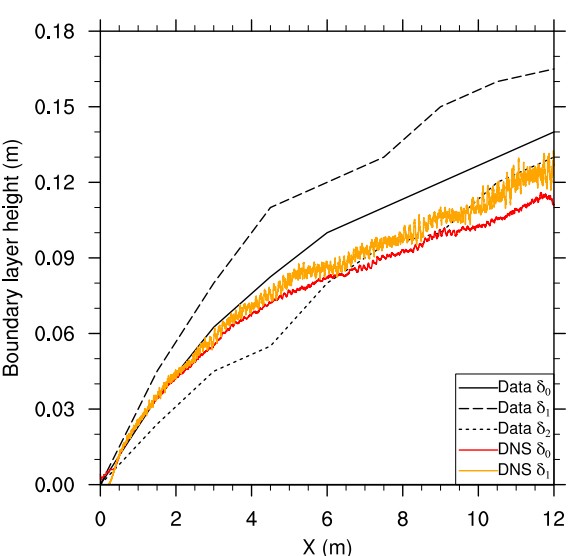

**Figure 11.** *Boundary layer height along the x-axis in stationary regime.*





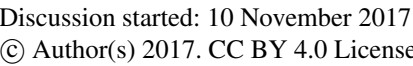

(a) x=1.5 m        (b) x=3 m        (c) x=4.5 m

(d) x=6 m        (e) x=7.5 m        (f) x=9 m

(g) x=10.5 m        (h) x=12 m

**Figure 12.** *Vertical profiles of mean velocities $(m.s^{-1})$ for various values of $x$ (m).*



(a)  x=1.5 m

(b)  x=3 m

(c)  x=4.5 m

(d)  x=6 m

(e)  x=7.5 m

(f)  x=9 m

(g)  x=10.5 m

(h)  x=12 m

**Figure 13.** *Vertical profiles of mean turbulence intensity (%) for various values of x (m).*



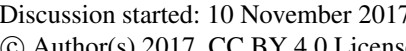


(a)  x=1.5 m

(b)  x=3 m

(c)  x=4.5 m

(d)  x=6 m

(e)  x=7.5 m

(f)  x=9 m

(g)  x=10.5 m

(h)  x=12 m

**Figure 14.** *Logarithmic regression of mean velocities vertical profiles for various values of x.*





|  | Length | Width | Height | Velocity | Viscosity |
|---|---|---|---|---|---|
|  | $L$ (m) | $W$ (m) | $H$ (m) | $U$ (m s$^{-1}$) | $\nu$ (m$^2$ s$^{-1}$) |
| Laboratory | 12 | 3 | 1 | 0.25 | $1.10^{-6}$ |
| Atmosphere | 19200 | 4800 | 1600 | 10 | 0.064 |

**Table 1.** *Values of the parameters characterising the flow at the scale of the laboratory and in the atmosphere.*

|  | Dimensions | Horizontal | Vertical | Time step | Duration |
|---|---|---|---|---|---|
|  | $L$ x $W$ x $H$ (m) | resolution (m) | resolution (m) | (s) | (s) |
| Laboratory scale | 12.5 x 0.2 x 0.6 | $10^{-3}$ | $10^{-3}$ to $10^{-2}$ | $5.10^{-3}$ | 83 |
| Atmospheric scale | 20000 x 320 x 960 | 1.6 | 1.6 to 16 | 0.2 | 3320 |

**Table 2.** *Main characteristics of **DNS** at the scale and dimensions of the laboratory and those of the atmosphere.*

| X (m) | 1.5 | 3 | 4.5 | 6 | 7.5 | 9 | 10.5 | 12 |
|---|---|---|---|---|---|---|---|---|
| Data | 0.8 | 1.4 | 1.1 | 1.1 | 1.0 | 1.0 | 1.1 | 0.9 |
| DNS | 1.1 | 1.1 | 1.0 | 1.0 | 1.0 | 1.0 | 1.0 | 1.0 |

**Table 3.** *Values of $u*$ ($10^{-2}$ m) for the experimental flow in the laboratory and for **DNS**.*

|  | X (m) | 1.5 | 3 | 4.5 | 6 | 7.5 | 9 | 10.5 | 12 |
|---|---|---|---|---|---|---|---|---|---|
| Data | $\widehat{u*}$ (cm/s) | 1.5 | 2 | 1.2 | 1.4 | 1.2 | 1.0 | 1.2 | 1.2 |
|  | $\widehat{z_0}$ ($10^{-2}$ mm) | 0.8 | 12 | 0.8 | 4.9 | 1.6 | 0.7 | 2.3 | 2.2 |
| DNS | $\widehat{u*}$ | 1.3 | 1.2 | 1.3 | 1.4 | 1.2 | 1.2 | 1.2 | 1.2 |
|  | $\widehat{z_0}$ ($10^{-2}$ mm) | 2.4 | 2.4 | 3.3 | 4.1 | 3.0 | 2.4 | 3.1 | 3.1 |

**Table 4.** *Values of $\widehat{u*}$ and $\widehat{z_0}$ deduced from the logarithmic regressions of the observed and simulated profiles of mean velocity $\overline{u}(z)$ (figure 14).*