# Peer review of "Adaptation of the meteorological model Meso-NH to laboratory experiments: implementations and validation"

_Geoscientific Model Development, 2017_

## Short Comment (SC1) · 16 Nov 2017

A. Kerkweg

kerkweg@uni-bonn.de

Dear authors,

In my role as Executive editor of GMD, I would like to bring to your attention our Editorial version 1.1:

http://www.geosci-model-dev.net/8/3487/2015/gmd-8-3487-2015.html

This highlights some requirements of papers published in GMD, which is also available on the GMD website in the 'Manuscript Types' section:

http://www.geoscientific-model-development.net/submission/manuscript_types.html

In particular, please note that for your paper, the following requirements have not been met in the Discussions paper:

- "The main paper must give the model name and version number (or other unique identifier) in the title."

- "All papers must include a section, at the end of the paper, entitled 'Code availability'. Here, either instructions for obtaining the code, or the reasons why the code is not available should be clearly stated. It is preferred for the code to be uploaded as a supplement or to be made available at a data repository with an associated DOI (digital object identifier) for the exact model version described in the paper. Alternatively, for established models, there may be an existing means of accessing the code through a particular system. In this case, there must exist a means of permanently accessing the precise model version described in the paper. In some cases, authors may prefer to put models on their own website, or to act as a point of contact for obtaining the code. Given the impermanence of websites and email addresses, this is not encouraged, and authors should consider improving the availability with a more permanent arrangement. After the paper is accepted the model archive should be updated to include a link to the GMD paper."

Note, that the exact Code version described in this article should be permanently accessible. Thus please consider to make the exact version, your article refers to, available via a permanent archive providing a DOI (e.g. Zenodo). Additionally, please add a version number, uniquely identifying this version, to the title of your article upon submission of the revised manuscript.

Yours,

Astrid Kerkweg

---

## Referee Comment (RC1) · Anonymous Referee #1 · 22 Nov 2017

Review of "*Adaptation of the Metreological model Meso-NH to laboratory experiments: Implementations and validation.* "

by J. Colin, C. Lac, V. Masson and A. Paci.

The manuscript presents an update to the well known Meso-NH model than enables it to run DNS. This is an interesting development and the details and testing deserve to published. Not only for the Meso-NH user base, but also others whom may wish to attempt a similar approach. First, the authors motivate their work by expressing their wish to explicitly resolve the turbulent motions in lab(-sized) experiments. Second, the two changes that were made to the code are presented: (1.) Including a viscous diffusion tendency term and (2.) the implementation of a Dirichlet-type bottom boundary condition for the momentum components. Third, 1D and 2D testcases are run to check the implementations of the aforementioned steps and finally, a comparison of the model results is made with the measurements obtained from a (3D turbulent) water tank experiment.

I agree with the authors that the development of atmospheric models can benefit from lab-experimental results. This concept is what attracted me to this study.

However, much of the content, arguments, and analysis presented in this manuscript would require very substantial revisions before I would consider the manuscript suitable for publication. Here is why:

**Major issues:**

**The Motivation**
What is de added value of the presented efforts compared to existing DNS codes? Is it the easy inclusion of the atmospheric physics modules that are already present in Meso-NH?

**The Implementation**
Eventough an entire section is devoted to it, the manuscript remains unclear on how the viscous diffusion term is actually implemented. I would like to see a more detailed description so that future readers will be able to reproduce the steps taken by the authors. (formulation with stencils etc). E.g. The fact that only one layer of ghost cells is defined suggest that the authors have opted for a second-order-accurate formulation. I would like to see this more explicit. Instead, section 2.2 reports on some general aspects of DNS and the authors' personal interpretation of that. Maybe the manuscript could do without the narrative?

**The Tests**
For all test cases, the setup seems rather arbitrary in terms of the chosen scales etc. A non-dimensional formulation of the problems would greatly help with the interpretation of the scales and results. I do not feel that bringing it to a "atmospheric scale" (that is apparently 4750m?) is very helpful as the tests bare no resemblance with the real atmosphere anyway.

The desired accuracy that the authors assume to be a low enough threshold is highly debatable and not motivated in the text. Often the authors even resort to statements as "Very close" to describe the comparison between the numerical results and the theoretical solutions. I find this is not very satisfactory.

Sect. 3.1.
The validation of the implementation of the diffusion-tendency term rather unconvincing. First, the test results are probably (I checked that this is true for atleast my personal favorite diffusion solver)

very sensitive to the way the boundary conditions are implemented and what they were chosen to be. This seems in contrast with the fact that this case was presented to be a test for the diffusion tendencies only. Furthermore, in order to upgrade this test to more of a validation-type analysis I would urge the authors to study the **spatial convergence properties** of their implementation for a **3D diffusion problem** (e.g. Gaussian pulse or a 3D extension of their periodic function on a triply periodic domain). Note that it is not obvious for me that a given order of spatial accuracy for a 1D problem is (naively) inherited by higher dimensional simulations.

Sect. 3.2 and 3.3.
I have similar objections to the current validation of the implementation of the new-boundary conditions with the first Stokes problem. It is clear that the analytical and numerically obtained solutions are rescaled versions of each other. Please use proper scaling (non-dimensional) for the analysis of the results. Provided that the test cases are suitable, a conclusion on the order of the spatial convergence rate is the only reasonable way to validate the implementations.

Finally, I think the quality of the work would greatly improve if the authors show that they are able to reproduce the results of the turbulent channel flow of Moser et al. (1999), these results have proven to provide excellent DNS benchmark results.

Ref: http://turbulence.ices.utexas.edu/MKM_1999.html

However, I can also understand that validation with the waterflume data could be a more attractive option. But that would require the quality of that part of the manuscript to be raised considerably.

**The lab experiment part**
As elegant as the Pi-Buckingham theorem may seem, it warrants careful consideration of the selection of parameters that define the flow. Here an issue arrises for me: It is not obvious to me why the gravity acceleration (g) is chosen to be part of the list of parameters that define the problem. The turbulence in the flume is neutrally stratified right? I can only imagine it has an effect on the surface water-air interracial waves. These are not resolved in the simulation, correct? Therefore I find the introduction of the Froude number confusing.

It not clear how the Reynolds number (25000) is computed (Lambda = ?). This also leaves open many question with regards to if the Kolmogorov scale is reasonably estimated.

The authors mention that the simulation is performed without any parameterization (L14p13). This is not true, the effect of the lateral walls, top wall (roof) the inflow and the outflow are (crude) parameterizations of the real experimental setup. Also the authors never address the actual implementations nor the effect of these parameterization choices.

L5-10 P14: The analysis where the iso surfaces of |U| are said to indicate isotropic turbulence and therefore prove that the flow must be accurately resolved over the whole spectrum is debatable on many levels.
- I have no clue what features of the graphs would be clear indicators of isotropy. I can only identify the presence of chaotic/erratic/irregular features by eye.
- What type of isotropy am I actually looking for? The most obvious feature I can spot is that |U| increases with height (i.e. an anisotropic feature).
- Assuming that there are hints of isotropy in the graphs, How do I know if they are indeed correct?
- Given the presence of a correct (isotropic) inertial subrange. How can I know if the smallest viscous lengthscales, that are also part of the spectrum, are resolved correctly?

A quantitative spectral analysis would be of great value here. I'd love to see the -5/3 scaling of the E(k) spectrum and **a consistent viscous range**. It is only conjecture, but at this moment I have strong doubts that the chosen grid spacing would be low enough for this purpose. I think it is required that I am proven wrong here.

L3p14, I have seen no results that indicate a fully resolved DNS. Please be more critical to the chosen benchmarks and the results. None of them prove that the inclusion of the diffusion term actually did something relevant for the flow evolution or has influenced the presented statistics. The only presented result that could be indicative of the of viscosity is the "law off the wall" analysis corresponding to Fig. 14. It worries me that there appears to be no resolved so-called viscous sublayer.
 See: https://en.wikipedia.org/wiki/Law_of_the_wall
Also it would greatly help interpretation of the profiles if they were presented in wall units.

Sect. 4.3 Reports a "rigid-roof" boundary condition. I am not familiar with these type of boundaries. Please explain what it means.

Also sect 4.3 does not present enough information on how the outflow boundary condition is implemented. In general, any outflow condition is not able to accurately describe the turbulence near that model boundary. A naive Neumann condition for the velocity component and pressure fields typically destroys turbulent structures that arrive at the boundary. This makes the resolved flow close to the boundary unphysical. I would therefore propose that the authors omit the data that is obtained close to the outflow boundary. Or in any case to quantify how close 'too close' in this respect.

Did the authors do tests to check if the presented results are converged with respect to the chosen setup of domain size, grid resolution, boundary conditions?

It also remains unclear that the DNS approach has actually added something compared to the "default" LES approach of Meso-NH. At this moment I think that the presented results may as well be obtained with the adoption of an Eddy-diffusivity-type closure. Since it is pivotal for the authors' motivation to run a DNS, I'd like to be proven wrong.

**Conclusion:**
From my previous comments it is clear that I disagree with the statements that the presented results prove the DNS capabilities of Meso-NH for 3D turbulent flows.

L27P16, "*To our knowledge, this is the first time an atmospheric model is successfully run in DNS.*" Such a statement does not do justice to the vast amount of atmospheric literature that appeared in the past decades that does employ a DNS strategy. It is even inconsistent with the introduction of this manuscript itself. In my opinion, these type of statements are not acceptable for publication.

**More issues:**

**0. Title:**
The title creates the impression that Meso-NH is adapted based on lab-experimental results. This is not the case. Rather it is extended to run in a DNS mode, motivated by a wish to explicitly resolve the lab experiments. Also, I disagree that the paper provides clear information on the implementations nor does it provide a validation (only tests).

**Abstract**

l3, The viscous diffusive fluxes in a LES are typically not simply neglected, rather their effect on the flow is parameterized. e.g. The dissipation of kinetic energy.

l7. The "very high resolution" claim is subjective. In the text it is approximated to be 20 times the viscous length scale, and hence, can also be considered to be coarse, as it operates at the limit (at best) of what a DNS should do.

**1. Introduction**

In the comparison between LES, DNS and lab experiment efforts to model the atmospheric boundary layer:

I think an important part for the motivation to use DNS in atmospheric research is missing. In fact all of the three aforementioned methods assume that it is possible to represent the enormous range of lengthscales present in a typical atmospheric turbulent flow with a much lower degree of (represented) scale separation. Either by employing closures (LES) or a reduced Reynolds Number, all method rely on the assumption that there is a presence of a large enough intertial subrange that displays a (down-scale) cascade and that the smallest scales do not directly influence the larger scales that dominate the overall dynamics.

J. Eggels et al (1994) presented very relevant work in JFM under the title: Fully developed turbulent pipe flow: A comparison between direct numerical simulation and experiment
Please consider to place the work presented in this manuscript in its context.

L35 p3. Expresses interest in the stable boundary layer. The (DNS-based) works of Nieuwstadt (2005), C. Ansorge & Mellado (2014), Donda et al. (2015), Van Hooijdonk et al. (2017), may be helpful references for the reader interested in this topic:

Nieuwstadt (2005): *Direct Numerical Simulation of Stable Channel Flow at Large Stability*
Ansorge & Mellado (2014) *Global Intermittency and collapsing turbulence in the stratified planetary boundary layer.*
Donda et al. (2015) *Collapse of turbulence in stable stratified channel flow: A transient phenomenon*
Van Hooijdonk et al (2017). *Early warning signals in the stable boundary layer: A model study*

**2. Imprementation**

To my knowledge, Meso-NH in LES mode works with an eddy-diffusivity-type closure. Apart from evaluation the eddy-diffusivity, it seems (implementation wise) very similar to how it affects the tendency terms compared to using a fixed-viscosity in the DNS. Why have the authors chosen to implement an additional viscous diffusion term? Or did they just override the turbulence-model and plugged-in a constant diffusivity? If no, why not?

Eq 10 and 11 report how the ghost cell values for the horizontal windcomponents are defined in an identical way. Do other fields get the same treatment? Like the Pressure, or potential temperature?

I would expect that the the 5th order WENO-advection-scheme would require atleast two layers of boundary-ghost cells (also for w). Yet the authors define only one layer? Also the authors seem to have opted for a second-order accurate definition of the ghost cell values. Please note this explicitly in the text.

**3. test cases**

I do not see why the diffusion tendency is tested with the U, p and Theta field. The inclusion of variables for momentum, pressure and potential temperature should not do anything in this case. It gives the false impression that there are all kinds of dimensionless parameters to be identified that define the problem. Whereas for a 1D-case, an analytical expression can be derived that has no explicit dependence on any of these parameters. Please consider to present the results in terms of a normalized species concentration.

The plotted initialized solution in fig. 2 is not consistent with eq. 12

Also eq 12: Capital Pi is introduced and not explained in the text. Later is appears to be replaced with lowercase pi.

The vector **x** in eq. 12 Should maybe read **e_x** to indicate that it is an normalized vector? In any case, please explain more clearly the intended meaning of the used symbols.

Please do not use subjective terms as 'very close' or similar statements for these simple numerical problems. I find the results wildly inaccurate.

**4. Experiment**

The Reynolds number is sometimes written with subscript ($R_e$) and sometimes not.

The Kolmogorov lengthscale is estimated to be a single value. I expect it would not vary throughout the domain? So what does the calculated Kolmogorov scale actually represent here? (a BL-Mean? A minimum?)

The text and numbers in fig. 9 are not legible.

It seems like the 8 panels in fig 12 and 14 display the same data? It that necessary?

How was the vertical gridstreching defined exactly and on what a priori knowledge was this based on?

L30p11 Please define turbulence intensity (I) at the first usage and not over 2 pages later.

L29P13 Adding a white noise for the velocity components seems to be inconsistent with Eq 2. Are there references I can read that show that this is a good idea to represent the inflow (or outflow)?

**5. Conclusions:**

I agree with the last section of the conclusions (starting with the sentence citing Stiperski et al. (2017)), that the DNS capabilities may well be used for these interesting purposes. However, they are hardly conclusions from the presented work.

---

## Referee Comment (RC2) · Anonymous Referee #2 · 6 Dec 2017

General comments

This paper presents an adaptation of the model Meso-NH to run Direct Numerical Simulations (DNS). The authors have introduced two modifications to the code: the addition of the viscous term and the non-slip boundary condition. The code is then tested against exact solutions of the Navier-Stokes equations and against laboratory experiments.

I think that the main idea of the paper is really good: to use a meteorological model in DNS mode in order to, for example, test subgrid parameterizations can be very useful. These experiments can be really helpful as resolution keep growing, and parameter-

ized processes, as turbulence or convection, are getting more resolved

However, I also think that the paper needs substantial improvement before being considered for publication: the test cases are either too simple or too complicated for testing the code, and the paper is often not rigorous enough. Besides, if the authors intend to use the model for complementing experiments in laboratories, they should provide convergence studies in order to be able to compare the results with other DNS studies. For these reasons I recommend major revisions.

Specific comments

1) In section 3 the authors show that Meso-SH recovers the analytic solutions of pure laminar flows. These tests show that the equations are properly coded, but they do not provide any extra information. I think that showing one single example is sufficient. It would be more useful to show convergence studies, in order to know which resolution is necessary with this numerical implementation. This would allow the comparison with other DNS codes. Also, the diffusion of a 2D-Gaussian field is a more common and challenging test for this kind of studies.

2) While I understand that the authors want to compare their simulations with the results of experiments of the same research center in section 4, this is not a good choice for validating the code. My reasons are the following. First, the DNS cannot simulate some features of the real flume, like the rough wall or the top open-boundary. Second, the data in the experiment are insufficient to validate the code close to the near-wall region, which is the most challenging part for boundary layer flows. Also, entrance effects in the flume seem to be very strong when compared with other experiments in wind tunnels, which makes difficult the comparison (therefore the different definitions of \delta). Third, simulating large Reynolds numbers like in the flume installation is quite challenging, and I do not think that the authors have the resources for that. In wall units, the vertical grid spacing can be estimated as delta $z+ = 10$, which is probably insufficient for boundary layer flows. All these factors might explain the differences

between experiment and simulations, but the validation of the code is not satisfactory. I would suggest the authors to compare their results with other DNS codes in channel or boundary layer flows, and/or with other experiments in smaller facilities. The classical references for boundary layer and channel flows, Spalart (1988) and Kim et al. (1987), are still good but probably more modern DNS/experiments could also be included.

3) The approximation of using a high viscosity in DNS to compare with meteorological flows in section 2.2 should be better discussed. My current experience is that many meteorologists still distrust DNS, and one should be very careful with this kind of comments. The high-viscosity approximation is only valid if Reynolds-number independency, and therefore enough scale separation, is achieved (see Pope). This is the case with turbulent mixing when $Re_\lambda \sim 50$ (Dimotakis 2005) for neutral stratification. However, even for very high $Re_\lambda$, the high-viscosity approximation can fail in regions where small eddies or viscous effects are important. These are the cases of the flow close to the wall, or with a strong stratification. Reynolds-number independency should be always checked with simulations with different Reynolds numbers, and not taken by granted. This is also the case for your boundary-layer simulation in section 4.

4) The authors should use non-dimensional numbers when describing the experiments in section 3, as it is standard in the fluid-dynamics literature. This makes easier the comparison with other experiments. For example, when writing the problem with non-dimensional numbers the three cases in section 3.2.2 are the same; they only differ in the time stepping. This comment is not against providing some reference length and time scales, which can be useful to some readers.

5) In section 4 the authors state which non-dimensional numbers are relevant for the experiment, but they do not discuss them. Please use the non-dimensional numbers to discuss which effects are relevant. The Froude number should be used to discuss if the free surface of the flume (which is not possible to simulate with the current code) is relevant. The aspect ratio d2 should be be used to discuss the width-effects. The length

of the flume could be use to discuss finite length effects. Discuss also the Reynolds number in the code and in the simulations.

6) Define the Komogorov length scale from the dissipation rate as it is commonly done in fluid dynamics literature. There is not point in providing only an approximation when you have all the data from the DNS.

7) Use wall units when presenting results in figures 14. Also it is more usual to fit \kappa and not ustar, which can be taken from measurements/simulations. You should also show the velocities close to the wall as they approach zero, and compare them to some boundary-layer reference (see above).

8) The roughness-length model cannot be applied to the simulations with non-slip boundary conditions, as done in page 15. The non-slip condition creates a viscous layer close to the wall, which is equivalent to the roughness layer but it is not the same. Using the proper viscosity and the right resolution is critical to get the right flow in the viscous layer. The flow close to the wall determines the surface drag u*, and therefore also influences the mean flow. It also determines the constant $B \sim 5.4$ from the Prandtl log law (see Pope page 274), which is independent from the roughness length. It is difficult to determine if the flow in the experimental facility close to the wall is dominated by roughness (there is no data of the roughness length), which makes difficult to compare with simulations.

Typos and small comments

1) Large grid in abstract is vague, use proper dimensions. 2) Page 2, line 23. I would use "complement the experimental data" instead of complete. 3) Page 4, line 10. Gas constant R 4) Equation (3). \rho_ref is missing from the momentum term. 5) Page 4, line 10. In LES the dissipation of energy is often done by the turbulence subgrid scheme. 6) Page 7, line 28. Bracket missing. 7) Section 3. A reference from a textbook would be useful when discussing simple flows. 8) Page 10, line 20. Why resolution does not really matter? 9) What do you mean by false floor in the experimental flume?

10) Page 15, line 12: shear velocity. 11) Figures 8 and 9. State which field is plotted.

References

Dimotakis, P. E., 2005: Turbulent mixing in stratified fluids. Annu. Rev. Fluid Mech., 37, 329–356, doi:10.1146/ annurev.fluid.36.050802.122015.

Pope, S. B., 2000: Turbulent Flows. Cambridge University Press
* * *

---

## Referee Comment (RC3) · Anonymous Referee #3 · 8 Dec 2017

The authors modify the existing large-scale model Meso-NH by inclusion of an explicit diffusion term and a no-slip boundary condition. The modified model is tested against two analytical solutions of laminar flows and experimental results for the Blasius boundary layer.

The title and abstract of this paper fit the scope of Geoscientific model development; a problem with the manuscript is, however, that it does not stand up to the promises given in the beginning due to a number of methodical problems. While the paper is well-written and the authors convey their messages clearly, the work is insufficiently embedded into the ongoing scientific discussion on direct numerical simulation which

has achieved considerable progress over the past decade or so. Key references are missing and the numerical testing is not up to the standards that should be put up for such simulation. It remains unclear which equations are actually solved by the so-called DNS and how these equations relate to the actual physical problems the reviewers claim to investigate.

In the reviewer's opinion, the paper is definitely not publishable in its present state, and I recommend to the editor to **reject the manuscript in its present form**.

**1   Reasons for Rejection**

There is two particular reasons for which I cannot recommend the publication of this paper. Beyond this, there are also a large number of major issues (see next section) which altogether, justify rejection of this work in the reviewer's opinion.

**I)** The authors do not demonstrate the convergence properties and accuracy of their numerical algorithm; the approaches presented are insufficient in a number of ways (cf. Rogallo and Moin (1984); Moin and Mahesh (1998); Coleman and Sandberg (2010)):

- First, a vague reproduction of analytical solution with errors on the order of $10^{-2}$ does neither serve as a demonstration of consistency nor does it prove convergence. Hence, the title and corresponding statements in introduction and conclusion about a validation of the numerical method cannot withstand a rigorous scientific assessment.

- Second, if the authors wish to establish their code as a DNS code, the code must also withstand rigorous testing of a truly three-dimensional turbulent case, and not only be validate approximately against noisy experimental data from a Blasius boundary layer. One such example is the turbulent Ekman layer (where a vast literature exists including very exact predictions for bulk turbulent properties

such as the surface veering angle and wall friction as a function of the Reynolds number and the geostrophic wind; cf. Coleman et al. (1990); Deusebio et al. (2014); Mellado and Ansorge (2012)), but tests with open or closed channel flows are also possible (cf. Moser et al. (1999)). I wish to point out that these tests, which can be carried out on grids of the order of $128^3$ would only require a fraction of the numerical resources used in the present work.

- In section 3.2, there is not only an error, but there seems to be a bias (the numerical solution of the model gives a consistently higher boundary layer than the analytical one). This should be explained. Further, it must be demonstrated that this bias approaches zero when the resolution in time and space is increased. Else, the model may be solving an equation different from the actual physical problem that the authors claim it solves.

**II)** The authors claim to establish Meso-NH as a tool for direct numerical simulation. This is not the case. The exact version of the governing equations which are solved is unclear, so I would like to pose the question DNS *of what?* The reason is that the authors lament about height-dependent outer-layer wind profiles (p. 9, l. 29) for cases where, from a physical perspective, there is clearly no reason for a height-dependency looking at the governing equations of the physical problems the authors claim to solve. There must hence be a discrepancy between the governing equations of the physical problem and the equations actually solved by the model.

**2 Major Issues**

M1 While I am not familiar with the details of the measurement procedures presented here, I would like to point out that such noisy measurements are insufficient to calibrate a DNS tool and much better test cases are available in the literature. In

principle do agree that the comparison of DNS results to observational or laboratory data is not only very illustrative to demonstrate the benefit of the method but also has substantial methodical benefits over the comparison with analytical results for turbulent flow, for the present work, I would suggest to focus on an improved demonstration of the actual convergence properties of the adapted numerical method before resorting to comparison with experimental data. Moreover, the mere availability of a dataset from an on-site laboratory should step back behind the availability of data at higher quality; and there definitely are much better-controlled data available for a Blasius-type boundary layer.

M3 While, as mentioned above, there are fundamental issues and technical problems associated with the use such a complex model in the mode that the authors refer to as 'DNS', the present manuscript does not mention methodological, technical or computational advantages of doing so in comparison to the established approach to DNS where light-weight but highly optimized implementations of PDE solvers are used to obtain the respective solutions.

M4 On page 5 (lines 9-10), the authors state that *meteorological models do not take into account viscous dissipation*. This is not true in general – effects of viscous dissipation are implicitly taken into account by either assuming that they are small or as part of the physical parameterizations. Eventually, the turbulent mixing, which is definitely taken into account, is a consequence of viscous adhesion.

M5 The explanation of the concept of Reynolds-number independence (p. 5, l.10-20) must be motivated and introduced better.

– *at the end of the spectrum* – which end do you refer to?

– *dissipation of energy [. . . ]is then ensured by a numerical diffusion scheme* – this is in general not true. Numerical diffusion is an artifact of the numerical scheme chosen and can be minimized by choice of a proper numerical

scheme. Turbulence diffusion is an explicit representation of the turbulent mixing as a function of resolved variables. There is only a small class of LES which use the numerical diffusion as part of the turbulent mixing.

M6 The order-of-magnitude estimates on p. 6, l3-14 are inconsistent in accuracy and thus cannot be reproduced (I obtain $\eta = 0.24$ from your values and using single-digit accuracy, this corresponds to $\eta = 0.2$ instead of $\eta = 0.1$; to resort to . Further, I consider this textbook-knowledge which I think does not need to be reproduced such broadly in a scientific paper.

M7 *From this validation, we conclude that our implementation of the viscous term is correct.* This is not possible. First, you must check for convergence and accuracy (and not consistence only) to claim this. Second, a check with a flow in the other direction is also needed to claim this.

**3 Further comments**

p.8,l.25 it is not possible to describe a whole boundary layer analytically; for certain aspects of it, this claim may be hold up.

p.10,l3 *Above, the resolution ranges from 2 m to 10 m and it is equal to 50m in the upper levels.* Does that mean, there is a factor-five jump in vertical resolution?

**References**

J P Mellado and C Ansorge. Factorization of the Fourier transform of the pressure-Poisson equation using finite differences in colocated grids. *Bound-Z. Angew. Math. Mech.*, 92(5): 380–392, 2012.

G N Coleman and R D Sandberg. A Primer on Direct Numertical Simulation of Turbulence - Methods, Procedures and Guidelines. Technical Report AFM-09/01b, Aerodynamics & Flight Mechanics Research Group, School of Engineering Sciences, University of Southampton, SO17 1BJ, UK, Southampton, 2010.

G N Coleman, J H Ferziger, and P R Spalart. A Numerical Study of the Turbulent Ekman Layer. *J Fluid Mech*, 213:313–348, 1990.

E Deusebio, G Brethouwer, P Schlatter, and E Lindborg. A numerical study of the unstratified and stratified Ekman layer. *J Fluid Mech*, 755:672–704, August 2014.

P Moin and K Mahesh. Direct numerical simulation: A tool in turbulence research. *Annu Rev Fluid Mech*, 30:539–578, 1998.

R D Moser, J Kim, and N N Mansour. Direct numerical simulation of turbulent channel flow up to Re[sub $\tau$]=590. *Phys Fluids*, 11(4):943, 1999.

R S Rogallo and P Moin. Numerical Simulation of Turbulent Flows. *Annu Rev Fluid Mech*, 16 (1):99–137, January 1984.

---

## Author Comment (AC1) · 29 Jan 2018

We would like to thank all the reviewers for taking the time to provide such detailed comments and suggestions.

However, given the negative reviews, we decided to withdraw the manuscript.